# ✒️LANCE: Stress-testing Visual Models by Generating Language-guided Counterfactual Images

**Viraj Prabhu**    **Sriram Yenamandra**    **Prithvijit Chattopadhyay**    **Judy Hoffman**
Georgia Institute of Technology
{virajp,sriramy,prithvijit3,judy}@gatech.edu

## Abstract

We propose an automated algorithm to stress-test a trained visual model by generating language-guided counterfactual test images (LANCE). Our method leverages recent progress in large language modeling and text-based image editing to augment an IID test set with a suite of diverse, realistic, and challenging test images without altering model weights. We benchmark the performance of a diverse set of pretrained models on our generated data and observe significant and consistent performance drops. We further analyze model sensitivity across different types of edits, and demonstrate its applicability at surfacing previously unknown class-level model biases in ImageNet. Code: https://github.com/virajprabhu/lance.

## 1   Introduction

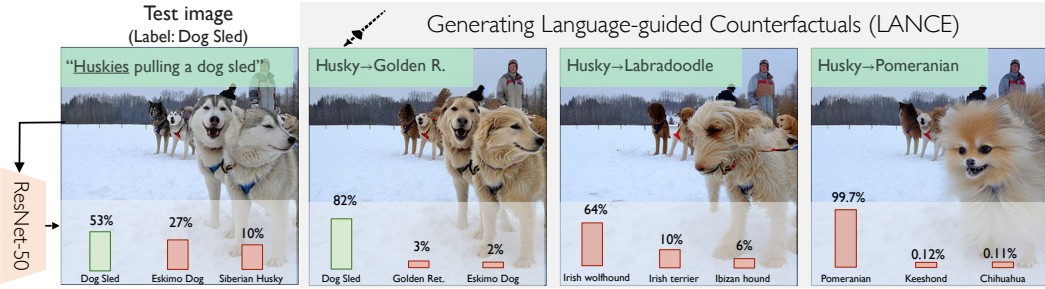

Figure 1: The predominant paradigm in computer vision is to benchmark trained models on IID test sets using aggregate metrics such as accuracy, which does not adequately vet models for deployment, *e.g.*for the test image above, while a trained ResNet-50 model [1] accurately predicts the ground truth label ("dog sled"), it is in truth *highly* sensitive to the dog *breed*. We propose LANCE, an automated method to surface model vulnerabilities across a diverse range of interventions by generating such counterfactual images using language guidance.

As deep visual models become ubiquitous in high-stakes applications, robust stress-testing assumes paramount importance. However, the traditional paradigm of evaluating performance on large-scale [2–4] IID test sets does not adequately vet models for deployment in the wild. First, such models are typically evaluated via aggregate metrics such as accuracy, IoU, or average precision [5], which treat all test samples equivalently and do not distinguish between error types. Further, such test sets do not adequately capture the "long-tail" [6] of the data distribution: rare concepts, unseen concepts, and the (combinatorial explosion) of their compositions [7].

To address this, a considerable number of recent efforts have sought to develop realistic benchmarks for *out-of-distribution* (OOD) evaluation [8–13]. However, while useful, such OOD benchmarks are typically *static* across models and time, rather than being curated to probe a specific instance of a trained model, which diminishes their utility.

37th Conference on Neural Information Processing Systems (NeurIPS 2023).

In this work, we eschew the traditional paradigm of static test sets and instead generate *dynamic test suites*. Specifically, we propose a technique that automatically generates a test suite of challenging but realistic *counterfactual* [14] examples to stress-test a given visual model. Our main insight is that while the scope of possible visual variation is vast, *language* can serve as a concise intermediate scaffold that captures salient visual features while abstracting away irrelevant detail. Further, while it is extremely challenging to run counterfactual queries on images directly, intervening on a discrete text representation is considerably easier. We call counterfactuals generated by our method Language-guided Counterfactual Images (LANCE).

Our method leverages pretrained foundation models for text-to-image synthesis [15–17] and large language modeling (LLM) [18], in combination with recent progress in text-based image editing [19, 20], to generate realistic counterfactual examples. It operates as follows: Given a test image and its ground truth label, we generate a caption conditioned on its label using an image captioning model [21]. We then use an LLM that has been fine-tuned for text editing [22] to generate realistic perturbations to a single concept at a time – caption subject, object, background, adjectives, or image domain (while leaving words corresponding to the ground truth category unchanged). Finally, we use a guided-diffusion model [19] with null-text inversion [20] to generate an edited counterfactual image conditioned on the original image and the perturbed prompt. We obtain a prediction from the trained model on both the original image and the generated counterfactual and compute the average drop in accuracy as a measure of the model's sensitivity to the given attribute.

We benchmark the performance of diverse models pretrained on ImageNet on our generated images, and find that performance drops more consistently and significantly than baseline approaches. Next, we demonstrate applications of our approach: in comparing the relative sensitivity of models to perturbations of different types, and in deriving actionable insights by isolating consistent class-level interventions that change model predictions.

## 2   Related Work

**Diagnosing deep visual models.** Several recent works have focused on the discovery of systematic failure modes in trained neural networks. Early works propose human-in-the-loop approaches based on annotating spurious features learned by sparse linear layers [23], adversarially robust models [24], and discovering a hyperplane corresponding to potentially biased attribute [25]. Some prior work also proposes fully automated techniques: by learning decision trees over misclassified instance features from an adversarially robust model [26] or targeted data collection [27]. Several recent approaches to this problem leverage multi-modal CLIP [28] embeddings, via error-aware mixture modeling [29], identifying failure directions in latent space [30] or learning the task directly on top of CLIP embeddings [31]. However, these works *rationalize* failures but don't close the loop by evaluating the predicted rationale. Further, shortcut learning [32] of spurious correlations can also lead to success but for the *wrong reasons*, but prior work only studies failure cases. In contrast, we propose a method that generates visual counterfactuals that can surface a diverse range of model biases that bidirectionally influence performance.

**Image Editing with generative models.** Using generative models to edit images has seen considerable work. Early efforts focused on specialized editing such as style transfer [33] or translating from one domain to another [34]. More recent work has performed editing in latent space of a model like StyleGAN [35–39]. Recently, pretrained text-to-image diffusion models have become the tool of choice for image editing [19, 20, 17]. While superior to GAN's at image synthesis [40], *targeted* editing using such models that modifies only a specific visual attribute while keeping the rest unchanged is non-trivial. Recent work has presented techniques based on prompt-to-prompt tuning [19], by targeted insertion of attention maps from cross-attention layers during the diffusion process. Follow-up work generalizes this approach to real images, with additional null-text inversion [20] or delta denoising scoring functions [41], which enables instruction-based editing of images [42] and 3D scenes [43]. We leverage this technique of prompt-to-prompt tuning with null-text inversion but rather than generic image editing focus on generating challenging counterfactual examples.

**Probing discriminative networks with generated counterfactuals.** Several works have attempted to use generative models to obtain explanations from visual models. A popular line of work generates minimal image perturbations that flip a model's prediction [44–47] (*e.g.*changing lip curvature to alter the prediction of a "smiling" classifier). Our work shares a similar goal but with a crucial distinction:

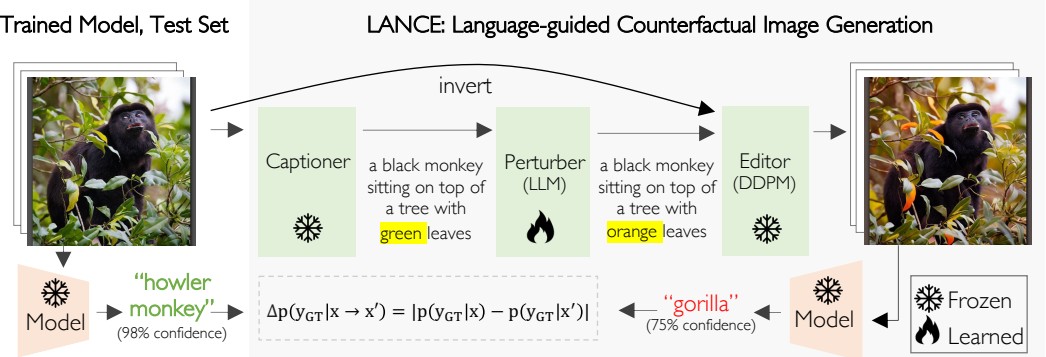

Figure 2: **Overview.** We propose a method to generate challenging counterfactual examples to stress-test a given visual model. Given a trained model and test set, LANCE generates a textual description (from a captioning model) and perturbed caption (using a large language model (LLM)), which is fed along with the original image to a text-to-image denoising diffusion probabilistic model (DDPM) to perform counterfactual editing. The process is repeated for multiple perturbations to generate a challenging test set. Finally, we ascertain model sensitivity to different factors of variation by reporting the change in the model accuracy over the corresponding counterfactual test set.

we seek to discover minimal perturbations to parts of the image *not* directly relevant to the task at hand (*e.g.*does changing hair color or perceived gender alter the smiling classifier's prediction?).

Closest to our work is Luo *et al.* [48], which determines model sensitivity to a set of user-defined text attributes by tuning a weighted combination of edit vectors in StyleGAN [35] latent space so as to flip the model prediction while maintaining global structure and semantics via additional losses. However, this method requires the user to enumerate all possible attributes of interest, may not generalize to more complex datasets, and requires optimizing several losses in conjunction. Our work also shares similar motivations as Li *et al.* [49], which uses diffusion models to generate ImageNet-E(diting), a robustness benchmark with varying backgrounds, object sizes, position, and directions. However, their benchmark is static across models, requires object masks per image to generate, and measures robustness to a constrained set of attribute changes. Our work is also related to Wiles *et al.* [50] which generates cluster-based error rationales, Vendrow *et al.* [51], which diagnoses failures by generating near-counterfactuals from human-generated text variations, and Dunlap *et al.* [52], which performs diffusion-based data augmentation. In contrast, we study bias *discovery* rather than mitigation, deriving causal insights into failures using a targeted editing algorithm, and support stress-testing across an unconstrained, automatically discovered set of attributes.

## 3 Language-guided Counterfactual Image Generation (LANCE)

**Overview.** We introduce LANCE, our algorithm for generating language-guided counterfactual images to stress-test a given visual model. Our key insight is to use language as a structured discrete scaffold to perform interventions on the model. LANCE does this by perturbing a text representation (caption) of an image to conditionally generate a counterfactual image (see Fig 2).

**Perturbed Captions.** We first use a pre-trained captioner (BLIP-2 [21]), to produce a text description for a given test image. We use beam search decoding with a minimum caption length of 20 words and a high repetition penalty to encourage descriptive captions with low redundancy (see Fig. 5 for examples). Our algorithm then edits the generated caption using a language model to generate variations that only change a *single* aspect at a time.

**Edited Images.** Next, we use a text-to-image latent diffusion model to generate a new image that reflects the text edit while remaining faithful to the original image in every other respect. We repeat this process for multiple perturbations to generate a challenging test set.

**Sensitivity Analysis.** Finally, we ascertain model sensitivity to different factors of variation by reporting the change in the model's accuracy over the corresponding counterfactual test set.

Next, we detail each step, beginning with describing the factors of visual variation that we stress-test against and the strategy we use to train a structured caption perturbed. Then, we describe our image

Table 1: **Perturbation dataset statistics.** We first programmatically collect a small dataset (0.6k samples) of 5 types of targeted caption perturbations for random MSCOCO [4] captions from GPT-3.5 turbo [22] (*top*). We then finetune LLAMA-7B [53] on this dataset to perform targeted perturbation, and use it to perturb image captions generated for ImageNet [2] (*bottom*).

| Generator | Sample Input | Edit Type (Count) | Sample Output |
|---|---|---|---|
| **GPT-3.5 turbo** (0.6k samples) | A bicycle replica with a clock as the front wheel | Subject (0.1k) | bicycle→{scooter, unicycle, ..., shopping cart} |
| | | Object (0.1k) | clock→{basketball, pizza, ..., globe} |
| | | Adjective (0.1k) | bicycle→{vintage, steampunk, ..., rusty} bicycle |
| | | Domain (0.1k) | a {painting, sculpture, ..., sketch} of a ... |
| | | Background (0.2k) | ... {on a stormy day, ..., parked outside a coffee shop} |
| **Finetuned LLAMA-7B** (35k samples) | a hockey puck sitting on top of a wooden table | Subject (6.2k) | hockey puck→{basketball, ..., lacrosse ball} |
| | | Object (4.8k) | wooden table→{marble table, ..., stone wall} |
| | | Adjective (7.1k) | hockey puck→ wooden, large, rustic hockey puck |
| | | Domain (3.4k) | a {painting, ..., 3D model} of a hockey puck... |
| | | Background (14.1k) | ... {in a busy hockey arena, ..., during a thunderstorm} |

editing methodology and the various checks and balances we insert to ensure that our end-to-end approach yields challenging yet realistic test examples.

## 3.1 Training a Structured Caption Perturber

To perform meaningful interventions, we train a structured caption perturber that generates targeted caption edits. Motivated by prior work in finetuning instruction-following large language models [22](LLMs), we seek to train a structured perturber LLM that can generate diverse and realistic caption edits that capture the long-tail of visual variation while still representing realistic scenarios.

**Selecting factors of variation.** To train our perturber model, we select five factors of visual variation against which we will stress-test. While the space of *potential* factors of visual variation is vast, image captions can constrain this search space by capturing salient visual concepts while abstracting away irrelevant detail. Concretely, we measure resilience to the following factors of variation:

- **Subject.** Modifications to the subject of an image caption (e.g., *a {man, dog, cat, ..., horse,}*) can stress-test a model's ability to recognize the ground truth category when it co-occurs with other subjects, including subjects that co-occur rarely (or never) with the ground truth in the training data.

- **Object.** Similarly, modifications to the object of an image caption (e.g., *a {table, chair, ..., bed}*) can stress-test a model's resilience to novel or unseen co-occurring concepts.

- **Background.** Modifications to the context of an image caption (e.g., *a {kitchen, bedroom, ..., living room}*) can stress-test a model's ability to generalize to different scenes, including diverse backgrounds and weather conditions.

- **Adjective.** Modifications to the adjective of an image caption (e.g., *a {red, blue, ..., green}*) can stress-test a model's ability to generalize to visual variations captured by object attributes.

- **Data domain.** Modifications to the data domain of an image caption (e.g., *a {painting, sketch, ..., sculpture}*) can stress-test a model's ability to generalize to different data distributions.

While by no means exhaustive, these perturbation types capture a large and representative set of visual variations. Further, we note that our approach is agnostic to this choice and can easily be extended to accommodate additional factors (e.g., camera angle, lighting, etc.). Having defined our perturbation set, we proceed to collect a dataset to train our structured perturbation model.

**Dataset collection.** We use GPT-3.5 turbo [18] to programmatically collect a small dataset of 0.6k caption perturbations spanning the aforementioned types. For instance, to edit an "adjective", we prompt the model with: *Generate all possible variations of the provided sentence by only adding or altering a single adjective or attribute.* (all prompts in appendix). We find that even with zero-shot prompting, we are able to acquire caption perturbations that modify only the desired factor of variation. We follow this procedure to generate perturbations for randomly selected captions from the MSCOCO dataset [4]. Table. 1 (*top*) highlights example inputs and perturbations. As seen, the dataset contains both simple edits (*e.g.*, *a bicycle →{red, blue, ..., green} bicycle*) as well as more complex ones (*e.g.*, *a bicycle →bicycle parked outside a coffee shop*), without requiring any human supervision whatsoever.

**Model training.** Next, we finetune an LLM on the dataset described above to perform targeted caption editing: specifically, we use a LLAMA-7B [53] LLM and perform LoRA (Low-Rank Adaptation) finetuning [54]: the original LLAMA-7B model is kept frozen, while the *change* in weight matrices $W \in \mathbb{R}^{d \times k}$ of the self-attention modules post-adaptation is restricted to have a low-rank decomposition $W_{ft} = W_{pt} + \Delta W = W_{pt} + AB$, where $A \in \mathbb{R}^{d \times r}$ and $B \in \mathbb{R}^{r \times k}$. Here the rank $r$ is kept low, resulting in very few trainable parameters in $A$ and $B$.

We use this finetuned model to perturb generated captions from ImageNet [2]. Table. 1 (*top*) shows examples of perturbations generated by this model. As seen, it is able to generate sensible and diverse edits corresponding to the edit type without changing the remainder of the caption.

**Caption editing: Checks and balances.** An undesirable type of edit would modify a word corresponding to the ground truth category label of the image. For instance, if the ground truth label of an image is 'dog', we do not want the model to generate a perturbation that changes the word 'dog' to 'cat'. To address this, we simply filter our caption edits wherein the edit is semantically similar (measured using a sentence BERT model [55]) to the ground truth category. Similarly, we also filter our captions where the edit is semantically similar to the original word or phrase being modified.

## 3.2 Counterfactual Image Generation

Having generated image captions and their perturbations, we proceed to generate counterfactual images conditioned on the original image and edited caption using a text-to-image latent diffusion model, specifically Stable Diffusion [16]. However, despite its remarkable ability at generating high-quality text-conditioned images, targeted editing using such models has historically been challenging, as changing even a single word in the text prompt could dramatically change the generated output image. Naturally, this is undesirable for our task, as we want to generate counterfactual images that are as similar as possible to the original image, while still reflecting the caption edit.

To address this, we leverage the recently proposed prompt-to-prompt [19] [1] image editing technique, which performs targeted injection of cross-attention maps that correspond to the caption edit for a subset of the denoising diffusion process. However, prompt-to-prompt is designed for generated images, and applying it to real images requires accurate image inversion to latent space.

We employ the recent null-text inversion technique proposed by Mokady *et al.* [20]. Concretely, for input image $\mathbf{x}$ with caption $c$, we perform DDIM inversion by setting the initial latent vector $z_0$ to the encoding of the image to be inverted $\mathbf{x}$, and run the diffusion process in the reverse direction [40] for $K$ timesteps $z_0 \rightarrow z_K$. Further, latent diffusion models employ classifier-free [59] guidance for text-conditioned generation, wherein the diffusion process is run twice, with text conditioning and unconditionally using a null-text token. To encourage accurate reconstruction of the original image, we follow Mokady *et al.* [20] to use the initial noisy diffusion trajectory (generated with a small guidance scale) as a pivot and update $\emptyset_k$, the null-text embedding at timestep $k$, to minimize a reconstruction mean square error between the predicted latent code $\hat{z}_k$ and the pivot $z_k$, using the default large guidance scale recommended for classifier-free guidance. This helps bring the backward diffusion trajectory close to the original image encoding $z_0$, and thus achieve faithful reconstruction. Let $S_{k-1}(\hat{z}_k, \emptyset_k, c)$ denote one step of deterministic DDIM sampling [60]. We optimize:

$$\min_{\emptyset_k} \|z_{k-1} - S_{k-1}(\hat{z}_k, \emptyset_k, c)\|_2^2 \tag{1}$$

**Image editing: Checks and balances.** We find that image editing using prompt-to-prompt with null-text inversion is highly sensitive to a specific hyperparameter which controls the fraction of diffusion steps for which self-attention maps for the original image are injected. Let $f$ denote this fraction. The optimal value of $f$ varies according to the degree of change, with larger changes (say editing the background or weather) requiring a small value. In service of making our algorithm fully automated, we follow prior work [43] to automatically tune this hyperparameter: we sweep over a range of values and threshold based on a CLIP [28] directional similarity metric [61], which measures the consistency in the change across images and captions in embedding space. Let $E_I$ and $E_T$ denote the CLIP image and text encoders. The CLIP directional similarity criterion $\phi(.)$ is given by:

$$\phi(\mathbf{x}, \mathbf{x}', c, c') = 1 - \frac{(E_I(\mathbf{x}) - E_I(\mathbf{x}')) \cdot (E_T(c) - E_T(c'))}{|E_I(\mathbf{x}) - E_I(\mathbf{x}')| \cdot |E_T(c) - E_T(c')|} \tag{2}$$

---

[1] Several editing techniques have been proposed recently [56, 57, 41, 58], which may be equally suitable.

**Algorithm 1** Generating Language-guided Counterfactual Images

---

1: **Input:** Test set $\mathbf{T}=\{(\text{image } \mathbf{x}, \text{ label } y)\}_1^M$, model $\mathcal{M}$, image captioner $\mathcal{C}$, perturber LLM $\mathcal{P}$, latent diffusion model $\mathcal{L}$, maximum perturbations per image $N$, edit similarity threshold $\epsilon$, image quality threshold $\tau$
2: **Output:** Counterfactual test set $\mathbf{T}'$
3: $\mathbf{T}' \leftarrow \emptyset$                                                 ▷ Initialize counterfactual test set
4: $\mathcal{P} \leftarrow \{\text{Subject, Object, Background, Domain, Adjective}\}$       ▷ Define perturbation types
5: **for** $(\mathbf{x}, y) \in \mathbf{T}$ **do**
6:      $\mathbf{x}^{-1} \leftarrow \mathcal{L}(\mathbf{x}, \mathbf{c})$                  ▷ Invert image to latent space using diffusion model
7:      $c \leftarrow \mathcal{C}(\mathbf{x})$                                   ▷ Generate caption for input image
8:      **for** $i \in 1, .., N$ **do**
9:          $p \sim \mathcal{P}$                                     ▷ Sample perturbation type
10:          $c' \leftarrow \mathcal{P}(c|p)$                              ▷ Perform text edit
11:          **if** $\text{sim}(c', y) < \epsilon$ **then**            ▷ Ensure ground truth is unchanged
12:              $\mathbf{x}' \leftarrow \mathcal{L}(c' \,|\, \mathbf{x}^{-1})$        ▷ Generate counterfactual image, Sec. 3.2
13:              **if** $\phi(\mathbf{x}, \mathbf{x}', c, c') > \tau$ **then**       ▷ Ensure image quality, Eq. 2
14:                  $\mathbf{T}' \leftarrow \mathbf{T}' \cup (\mathbf{x}', y)$            ▷ Add to test set
15: **return** $\mathbf{T}'$

---

Finally, we ensure that the generated image is more similar to the edited rather than the original caption in CLIP's [28] embedding space. Algo. 1 details our full approach.

## 4 Experiments

In Section 4.1, we overview our experimental setup, describing the data, metrics, baselines, and implementation details used. Next, we present our results (Section 4.2), comparing the performance of a diverse set of pretrained models on the subset of the ImageNet test set, and on our generated counterfactual test sets. Additionally, we analyze the sensitivity of models to different types of edits and demonstrate the applicability of our method in deriving class-level insights into model bias.

### 4.1 Experimental Setup

**Dataset.** We evaluate LANCE on a subset of the ImageNet [2] validation set. Specifically, we study the 15 classes included in the Hard ImageNet benchmark [62]. Models trained on ImageNet have been shown to rely heavily on spurious features (*e.g.* context) to make predictions for these classes, which makes it an ideal testbed for our approach. We consider the original ImageNet validation sets for these 15 classes, with 50 images/class, as our base set. Further, testing our approach on ImageNet has a few other advantages: i) The dataset contains naturally occurring spurious correlations rather than manually generated ones (*e.g.* by introducing a dataset imbalance). ii) Testing our method on ImageNet images rather than constrained settings used in prior work *e.g.* celebrity faces in CelebA allows us to validate its effectiveness in more practical settings. iii) Finally, using ImageNet allows us to stress-test a wide range of pretrained models.

**Metrics.** We report the model's accuracy@k ($k \in \{1, 5\}$) over the original test set $\mathbf{T}$ and generated counterfactual test set $\mathbf{T}'$. For model $\mathcal{M}$, we are interested in understanding the *drop* in accuracy@k over the counterfactual test set, compared to the original test set. We define this metric as follows:

$$\Delta\text{acc@k} = \Big[\frac{1}{|\mathbf{T}'|} \sum_{(\mathbf{x}',y) \in \mathbf{T}'} \text{acc@k}(\mathcal{M}(\mathbf{x}'), y)\Big] - \Big[\frac{1}{|\mathbf{T}|} \sum_{(\mathbf{x},y) \in \mathbf{T}} \text{acc@k}(\mathcal{M}(\mathbf{x}), y)\Big] \quad (3)$$

However, it is possible for an intervention to alter model confidence without leading to a change in its final prediction. As a more fine-grained measure, we also report the absolute difference in predicted model confidence for the ground truth class over the original and counterfactual images. For a single instance, we define this as: $\Delta p(y_{GT}|\mathbf{x}) = |p(y_{GT}|\mathbf{x}) - p(y_{GT}|\mathbf{x}')|$

Finally, to evaluate realism we also report the FID [63] score of our generated image test sets and perplexity of the generated and perturbed captions under LLAMA-7B [53].

Table 2: **Results.** We evaluate diverse models pretrained on ImageNet-1K on four test sets: the original Hard ImageNet test set (reference), its reconstruction generated by our method (control), and the counterfactual test sets generated by a random perturbation baseline (Ours-Baseline), and our proposed structured perturbation strategy (Ours). Our methods generate progressively more challenging test sets as evidenced by the consistent performance drop compared to the original test set ($\Delta$acc@k, subscript).

| | Test Set | Type | acc@1 / acc@5 ($\uparrow$) | | | |
|---|---|---|---|---|---|---|
| | | | ResNet-50 [1] | ViT-B [67] | ConvNext [68] | CLIP [28] |
| 1 | Original | Reference | 79.86 / 95.83 | 85.42 / 97.74 | 85.42 / 98.26 | 56.25 / 86.98 |
| 2 | Reconstructed | Control | $79.86_{+0.0}$ / $96.18_{+0.4}$ | $85.42_{+0.0}$ / $97.57_{-0.2}$ | $85.24_{-0.2}$ / $98.09_{-0.2}$ | $55.90_{-0.4}$ / $87.67_{+0.7}$ |
| 3 | LANCE-R | Ours - Baseline | $78.81_{-1.0}$ / $92.25_{-3.6}$ | $81.91_{-3.5}$ / $95.61_{-2.1}$ | $85.79_{+0.4}$ / $94.83_{-3.4}$ | $50.13_{-6.1}$ / $83.46_{-3.5}$ |
| 4 | LANCE | Ours | $74.01_{-5.8}$ / $92.96_{-2.9}$ | $79.00_{-6.4}$ / $94.24_{-3.5}$ | $81.18_{-4.2}$ / $94.62_{-3.6}$ | $48.27_{-8.0}$ / $84.38_{-2.6}$ |

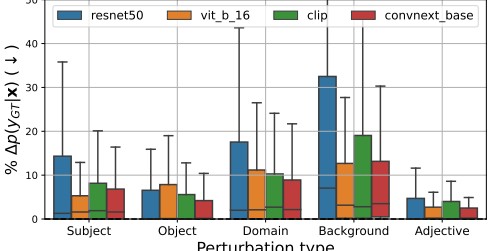

| Method | FID($\downarrow$) | Perplexity($\downarrow$) |
|---|---|---|
| Original | 0 | 17.1 |
| Reconstructed | 13.6 | 17.5 |
| LANCE-R | 51.6 | 18.3 |
| LANCE | 55.4 | 18.6 |

Figure 3: Sensitivity across perturbation types as measured by average $\Delta p(y_{GT}|\mathbf{x})$.

Figure 4: Evaluating quality of generated images (FID) and captions (perplexity).

**Implementation Details.** We use BLIP-2 (for image captioning), LLAMA-7B [53] (for structured caption perturbation) with LoRA finetuning [54], StableDiffusion [16] (for text-to-image generation), and pretrained models from TIMM [64]. We use PyTorch [65] for all experiments. For finetuning LLaMA-7B, we concatenate the query, key, and value matrices per layer and approximate the updates to the resulting weight matrix using a rank-8 decomposition. We finetune the model for 37.5k steps updating the weights every 32 steps. For more details see appendix.

**Baselines.** In the absence of prior empirical work for our experimental setting, we compare the performance of LANCE to the following:

i) **Original** (Reference): The original, unmodified ImageNet test set, as a point of reference.

i) **Reconstructed** (Control): The reconstruction of the original ImageNet test set, as a control set

iii) **LANCE-R** (Baseline, Ours): As a baseline, we design a simple *random* caption perturbation strategy that randomly masks out a word in the original caption and replaces it with a different word using a masked language model [66]. To ensure meaningful perturbations, we impose two additional constraints: the word being modified should not be a stop word (to minimize wasteful edits *e.g.* changing 'a' to 'the'), and that the CLIP [28] similarity between the generated image and the new word should exceed its similarity to the original word.

## 4.2 Results

**Across pretrained models, LANCE generates more challenging counterfactuals than baselines.** Our goal is to discover a set of example images that are especially challenging for a given model. Therefore, Table 2 reports top-1 and top-5 performance of multiple fixed models across two baseline test sets and our own (LANCE). We find that every model studied has significantly lower performance on our proposed generated test set. We compare this both to the standard ImageNet test set as well as a baseline variant of our method that uses random caption perturbations (LANCE-R). Finally, we verify that performance loss is not due to the image generation process by demonstrating that performance on a reconstructed test set is nearly identical to the original test set.

We provide example LANCE perturbations and generated images in Figure 5. You can find an example image for each perturbation type. In some cases, such as the domain edit (changing from

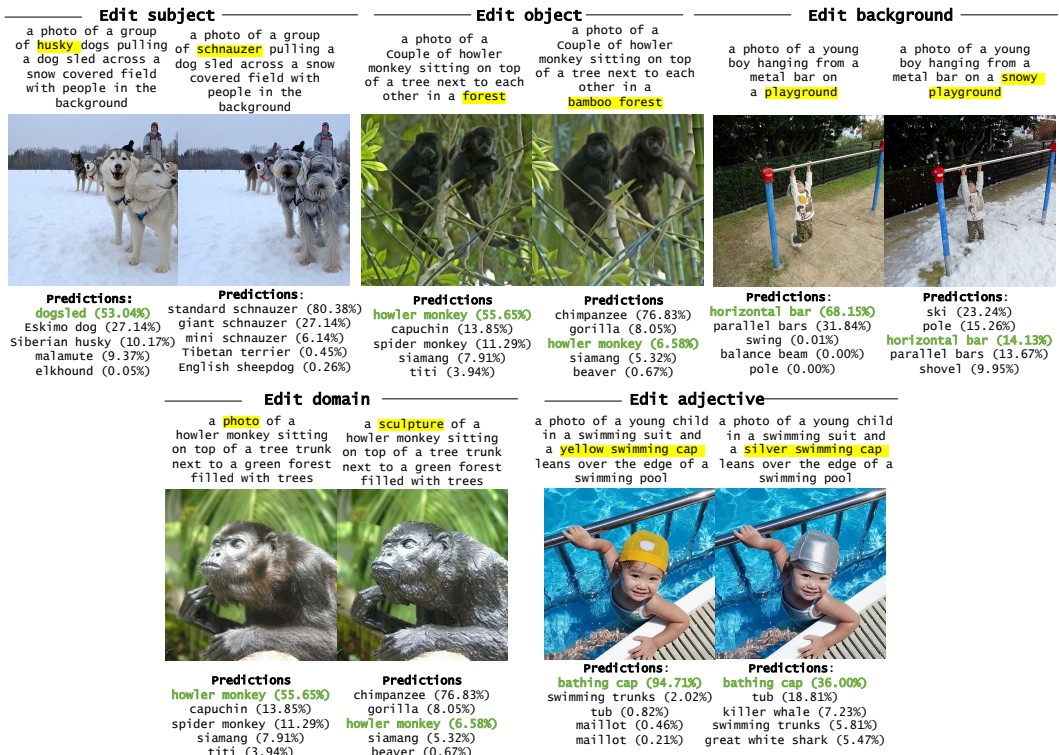

Figure 5: **Visualizing counterfactual images**. We visualize the counterfactual images generated by LANCE for images from the HardImageNet dataset. Each row corresponds to a specific edit type. Above each image, we display its generated caption, highlighting the original and edited word(s). Below each image, we display the top-5 classes predicted by a ResNet-50 model and the associated model confidence, coloring the ground truth class in green.

a photo to a sculpture) significant predictive changes result – from correctly identifying a howler monkey to incorrectly predicting a chimpanzee. As another example, the subject edit (changing the dogs from husky to schnauzer) prohibits the model from accurately predicting dogsled, conceivably because few schnauzers appeared pulling a sled in the training data.

**Perturbation sensitivity varies by perturbation type and model.** Now that we have confirmed that our generated test set provides challenging examples to assess our models, we may further analyze the impact of different types of perturbations. Figure 3 reports the relative change in predictive performance in response to different perturbation types. We find that changes to the background have the highest impact, followed by domain. Further, some models are more sensitive to LANCE images, with ResNet-50 [1] having the most sensitivity.

**LANCE can be used to derive class-level insights into model bias.** To do so, we compute the L1 distance between CLIP features for the original and edited words, and run K-Means clustering. We then visualize the clusters with the highest $\Delta p(y_{GT}|\mathbf{x})$, corresponding to high sensitivity to a given change. Figure 6 illustrates examples. As seen, our method can also be used to surface class-level failure modes to inform downstream mitigation strategies. Importantly, we stress that while some failure modes are corroborated by other diagnostic datasets [62], (*e.g.*using color-based context-cues to predict "howler monkey"), others go beyond the capabilities of conventional IID test set-based diagnosis (*e.g.*models relying on dog breed to predict "dog sled"; see Figs. 1, 11).

**Are generated images and perturbed captions realistic?** As we are *generating* a new test set, it is important to verify that any performance changes are not due to an impact on the realism of our images. In Table 4 we report the FID scores of our generated images and the perplexity scores of our generated captions and observe both to be within a reasonable range. Further, we reiterate that using our pipeline to reconstruct the original image results in a test set where each model measures near identical performance (Table 2).

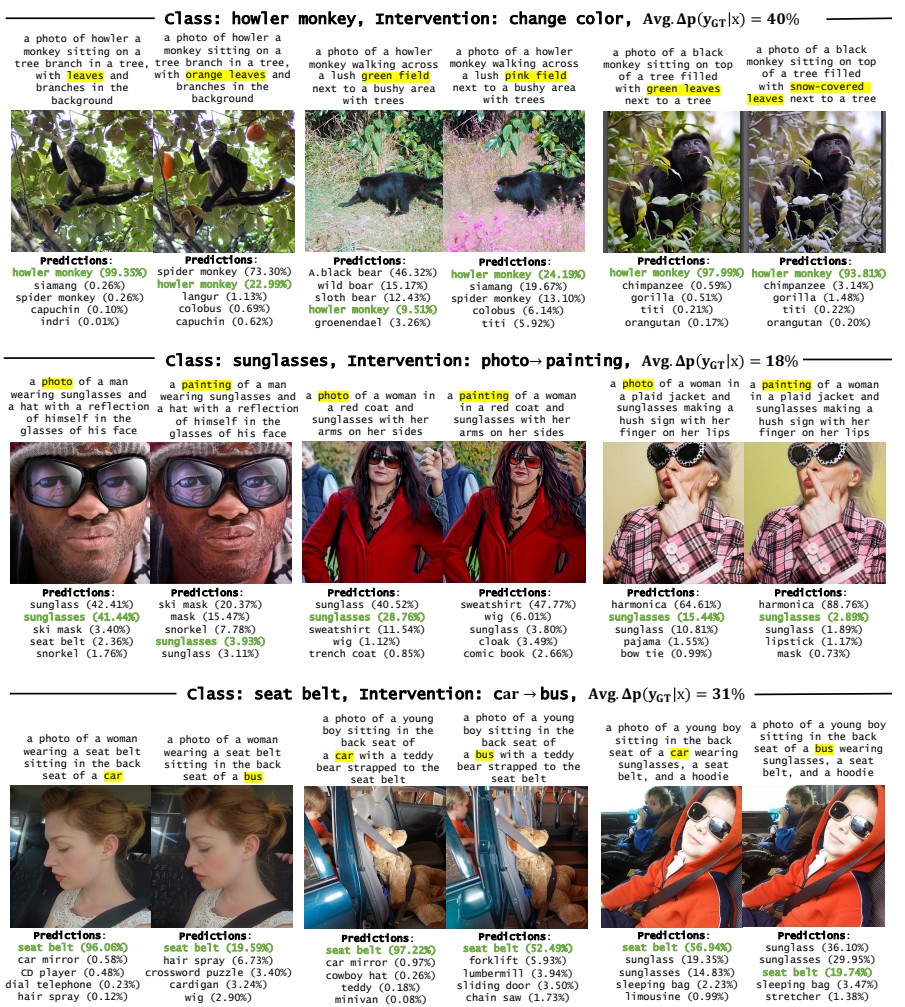

Figure 6: **Deriving class-level insights.** LANCE can be used to derive model bias at a per-class level by clustering text edits and visualizing clusters with the highest average $\Delta p(y_{GT}|\mathbf{x})$. Above we highlight insights for three classes with a ResNet-50 model: i) "howler monkey": high sensitivity to the background color ii) "sunglasses": high sensitivity to the data domain, and iii) seat belt: high sensitivity to the (barely perceptible) vehicle type.

Table 3: **Evaluating caption perturber.** LANCE performs successful edits without mutating labels.

| Metric | Subject (%) | Object (%) | Adjective (%) | Domain (%) | Background (%) | Overall (%) |
|---|---|---|---|---|---|---|
| Edit Success (%) | 92 | 84 | 82 | 98 | 92 | 89 |
| Filter prec. (%) / recall (%) | 99.0/97.0 | 100/96.0 | 92.9/98.9 | 100/100 | 94.1/100 | 97.2/98.4 |

**LANCE's checks & balances are highly effective.** Next, we evaluate the accuracy of the caption perturber at making the type of edit specified by the prompt (*e.g.* correctly changing only the domain for a domain edit). To do so, we manually validate a random subset of 500 captions and their perturbations (100 per type) generated by the LANCE. We observe an overall accuracy of 89%, with the object (84%) and adjective (82%) types achieving the lowest accuracies: we find that most failures for these types result from incorrect identification of the object or adjective in the sentence. We note that even in failure cases (*e.g.* changing an irrelevant word), the edits made are still reasonable.

Next, we evaluate the efficacy of our caption editing checks and balances. We first label the same subset of 500 examples by hand with the correct action (filter / no filter) and compare against LANCE. Across perturbation types, LANCE achieves both high precision and recall, with a low overall false positive (2.8%) and false negative (1.6%) rate. Of these, a majority of false negatives (edits that alter the ground truth that we fail to catch) are for the subject and object types: we find these typically

Table 4: **Human study.** LANCE-generated images score high on realism, edit success, and fidelity.

| Metric | Subject | Object | Adjective | Domain | Background | Overall |
|---|---|---|---|---|---|---|
| **Image Realism (1-5)** | 4.2±0.5 | 4.4±0.5 | 4.1±0.4 | 3.9±0.8 | 4.2±0.6 | 4.2±0.5 |
| **Edit Success (1-5)** | 3.8±0.4 | 3.6±0.5 | 4.1±0.5 | 3.7±0.5 | 3.3±0.4 | 3.7±0.4 |
| **Image Fidelity (1-5)** | 4.7±0.4 | 4.5±0.5 | 4.6±0.5 | 4.7±0.2 | 4.4±0.5 | 4.5±0.4 |
| **Label Consistency (%)** | 97.0±4.0 | 94.0±6.0 | 99.0±0 | 93.0±9.0 | 93.0±9.0 | 95.0±4.0 |
| **Ethical issues (%)** | 3.0±7.0 | 1.0±3.0 | 5.0±8.0 | 0±0 | 0±0 | 2.0±2.6 |

change the ground truth *inadvertently e.g.*for a "keyboard space bar" image of a typewriter, the edit typewriter→ painting erroneously passes our filter (paintings do not usually have space bars).

**Human evaluators validate the realism and efficacy of LANCE-generated edits.** We conduct a human evaluation along 5 axes: i) Image Realism (1-5, 5=best): How easy is it to tell that the counterfactual image was generated by AI? ii) Edit success (1-5, 5=best): Does the generated image correctly incorporate the edit? iii) Image fidelity (1-5, 5=best): Are all the changes made relevant to accomplishing the edit? iv) Label consistency (Yes/No): Is the original image label still plausible for the generated image?, and v) Ethical issues (Text input): Is the generated image objectionable, or raise ethical concerns around consent, privacy, stereotypes, demographics, etc.?

We collect responses from 15 external respondents for a random subset of 50 <image, ground truth, generated caption, perturbed caption, counterfactual image> tuples (10 per-perturbation type), and report mean and variance. We include a screenshot of our study interface in Fig. 13, and report results in Table 4. As seen, images generated by LANCE are rated to be high on realism (4.2/5 on average) and fidelity (4.5/5 on average), and slightly lower on edit quality (3.7/5 on average). Of these, we find that background edits score lowest on edit success, due to sometimes altering the wrong region of the image. Further, generated images have high label consistency (95%), verifying the efficacy of LANCE's checks and balances. Finally, ~2% of images are found to raise ethical concerns, which we further analyze below.

**Ethical Issues.** On inspection of the images marked as objectionable in our human study, we find that most arise from bias encoded in the generative model: *e.g.*for the edit woman→lifeguard, the generative model also alters the perceived gender and makes the individual muscular. Further, for people→athletes, the model also alters the person's perceived race. The model also sometimes imbues stereotypical definitions of subjective characteristics such as "stylish" (*e.g.*by adding makeup). To address this, we manually inspect the entire dataset of 781 images for similar issues and exclude 8 additional images. Going forward, we hope to ameliorate this by excluding subjective text edits and using text-to-image models with improved fairness [69] and steerability [57].

**Failure modes and limitations.** In Fig 12, we visualize LANCE's 5 most frequent failure modes: a) Label-incompatible text edit: *e.g.*black howler monkey→purple howler monkey (howler monkeys are almost always black or red). b) Poor quality image edit: Images with low realism, which are largely eliminated via our checks and balances. c) Image edit that inadvertently changes the ground truth: *e.g.*typewriter→stapler for the class "space bar" (staplers do not contain space bars). d) Inherited bias: *e.g.*changing perceived gender based on stereotypes. e) Meaningless caption edit: *e.g.*boy in back-seat driver in the back seat. Such failures can confound an observed performance drop but this is largely mitigated by their low overall incidence.

LANCE also has a few intrinsic limitations. Firstly, it leverages several large-scale pretrained models, each of which may possess its own inherent biases and failure modes, which is highly challenging to control for. Secondly, intervening on language may exclude more abstract perturbations that are not easily expressible in words. However, we envision that LANCE will directly benefit from the rapid ongoing progress in the generative capabilities [70, 53] and steerability [57] of image and text foundation models, paving the road towards robust stress-testing in high-stakes AI applications.

**Acknowledgements.** This work was supported in part by funding from NSF #2144194, Cisco, Google, and DARPA LwLL. We thank Simar Kareer for help with figures, members of the Hoffman Lab for project feedback, and the volunteers who participated in our human study.

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

# Appendix

## A    Dataset Details

**Assets and license.** All source images belong to the ImageNet dataset [2], which is distributed under a BSD-3 license that permits research and commercial use.

**Statistics.** We now provide an overview of the counterfactual test set generated via our method. After adding the checks and balances described in the main paper to ensure caption and image quality after editing, we approximately double the size of the HardImagenet dataset (from 750 to 1531). We provide a per-class breakdown of the generated samples in Figure 7: we find that our method is able to generate a larger number of counterfactuals for certain categories (*e.g.* "swimming cap") than others (*e.g.* "hockey puck"), presumably because one of two possible reasons: i) the structured caption perturber is able to generate several more plausible variations for certain classes, or ii) the checks and balances in our image editing pipeline filters out many more examples of certain classes due to not being of sufficiently high quality. We note that while we create this dataset as a proof-of-concept for our method, scaling it up to include additional classes and concepts is very straightforward, and simply requires finetuning the perturber on (a small number of) additional caption editing examples, which are relatively easy to collect.

## B    Per-class Analysis

In the main paper, we quantified performance in aggregate and per perturbation-type. We now investigate per-class performance over generated counterfactuals in Figure 8. We find that the counterfactuals generated for certain categories (*e.g.* "ski" or "swimming cap") are significantly more challenging across models than others.

**Do class-level insights generalize to the full dataset?** Recall that in Figure 3 of the main paper we derived class-level insights into model bias, for example finding that the accuracy of ResNet-50 models at recognizing "sunglasses" drops signficantly if the data domain is changed from "photo" to "painting". A natural question then is whether such insights generalize directly to other classes. In Figure 9, we apply this intervention to the entire HardImageNet dataset and report the per-class drop in top-1 accuracy when going from the original test set to the generated counterfactual test set. As seen, across all but one category ("baseball player"), performance drops, often significantly (*e.g.* "balance beam" and "hockey puck").

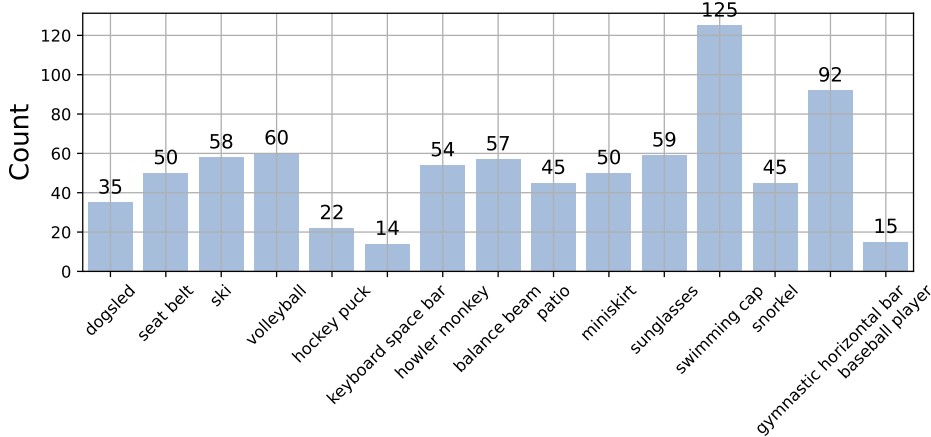

Figure 7: Label histogram of additional counterfactual images generated via LANCE.

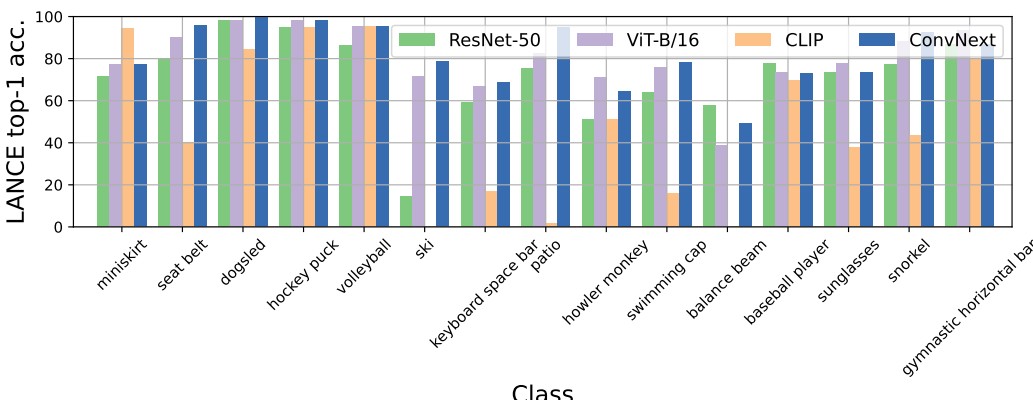

Figure 8: Per-class top-1 accuracy of trained models on the counterfactual images generated by LANCE on the HardImageNet [62] dataset.

## C  Additional Qualitative Results

We provide additional qualitative examples generated via LANCE in Figure 10. We demonstrate 3 examples for each perturbation type: subject, object, background, domain, and adjective. Some of these examples are particularly interesting *e.g.* that the model's confidence of there being a "balance beam" present goes down significantly if a "coach" is pictured doing a handstand (top row, *middle*). Similarly, the probability of the image containing a "patio" under the model goes down considerably if benches are replaced by bicycles (row 2, *left*). Similarly, the model can no longer recognize a "dogsled" if the weather conditions become misty (row 3, *left*). Finally, the model is much less confident in recognizing a "seatbelt" if the color of its buckle changes from silver to gold (bottom row, *middle*). We note that these are only image-level insights and require further investigation across images of a given class to draw conclusions of systematic model bias.

## D  List of Prompts

In Table 5 we provide an exhaustive list of the prompts we use for both GPT-3.5 turbo [18] and for finetuning LLAMA [53]. As seen, despite their simplicity our prompts are able to elicit the desired behavior from both models. We find that including a couple of examples in the prompt greatly improves performance.

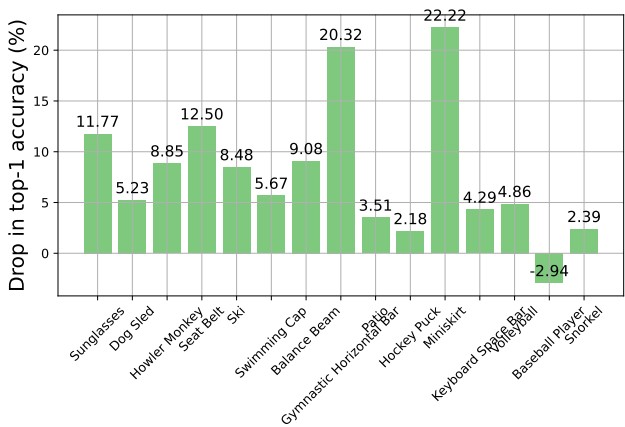

Figure 9: Per-class drop in top-1 accuracy of a trained ResNet-50 [1] model on the counterfactual images generated by LANCE for the *photo→painting* intervention.

# E    Analyzing Failure Modes

Despite the considerable number of checks and balances that we insert into our pipeline, while running it end-to-end automatically and at scale, we find that a few poor counterfactuals still fall through the cracks. We provide qualitative examples of representative failures modes in Figure 12, that we group into five categories:

i) **Inconsistent with GT.** LANCE sometimes generates edits that, while being reasonable in isolation, generates an image that is inconsistent with the ground truth label. For example, "howler monkeys" are always *black* or *blonde* in color, and so a *purple* howler monkey is statistically unlikely.

ii) **Meaningless caption edit.** In some cases, the structured perturber makes an edit that is visually meaningless, *e.g.* changing *photo* to *video*, while our pipeline is constrained to image generation.

iii) **Poor quality image edit.** Though we find this to occur very rarely due to our image quality filters, a few poor quality image edits are still retained *e.g.* accidentally cropping the limb of a person.

iv) **Changing GT.** A few edits inadvertently change the ground truth. For example, for the category "space bar", changing a *typewriter* to a *stapler* also ends up removing the space bar.

v) **Image generation bias.** Finally, we find that the latent diffusion model's inherent bias also sometimes seeps through, *e.g.* when changing an image of a *woman* to a *middle-aged woman* (ground truth category: "miniskirt"), the diffusion model also subtly changes the woman's attire to presumably be closer to attires commonly worn by middle-aged women in its training data, rather than changing physical age markers alone: in this case, the image cannot strictly be considered a counterfactual as multiple attributes are being intervened on.

# F    Implementation Details

We run all experiments on a single NVIDIA A40 GPU. All the models that we evaluate are trained with supervised learning on ImageNet-1K [2]. We include additional implementation details for hyperparameters used by LANCE for caption and image editing in Table 6.

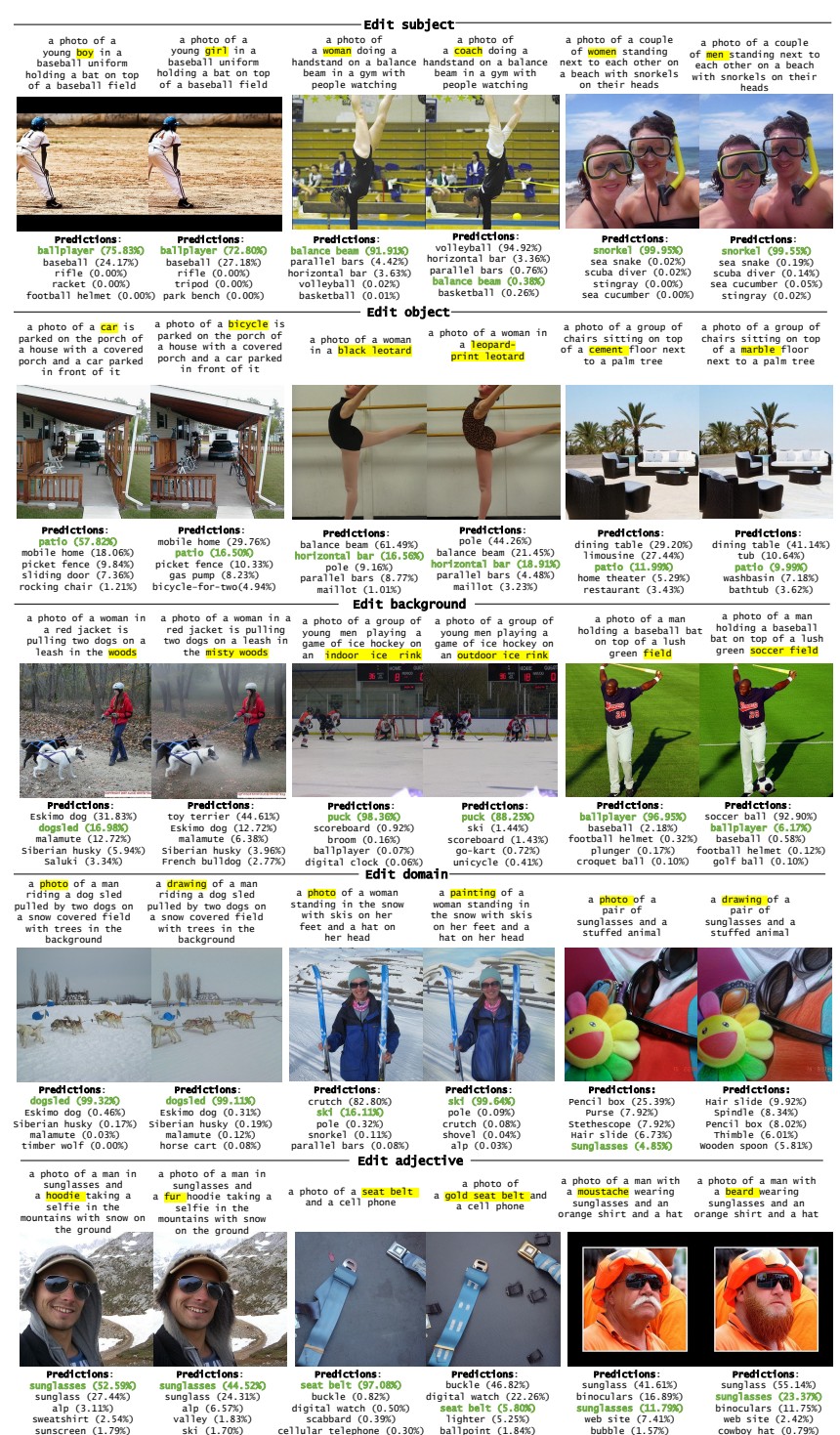

Figure 10: We visualize the counterfactual images generated by LANCE for images from the HardImageNet [62] dataset. Each row corresponds to a specific edit type. Above each image, we display its generated caption, highlighting the original and edited word(s). Below each image, we display the top-5 classes predicted by a ResNet-50 model and the associated model confidence, coloring the ground truth class in green.

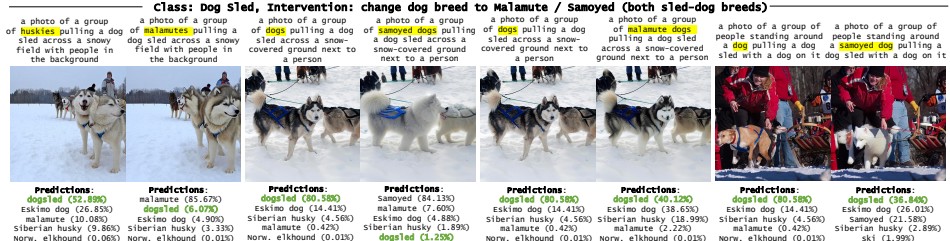

Figure 11: **Beyond conventional diagnosis.** LANCE can uncover nuanced model bias, *e.g.* predictive confidence for "dog sled" drops significantly even if the dog breed is changed to other popular sled dog breeds.

Table 5: Prompts used for programmatically collecting perturbations of captions using GPT-3.5 turbo [18]. We use the same prompts without examples as instructions when instruction fine-tuning LLAMA-7B [53] with LoRA [54]

| Perturbation type | Prompt |
|---|---|
| Edit subject | *Generate all possible variations by changing only the subject of the provided sentence. For example: Change "A man walking a dog" to "A woman walking a dog". <caption>* |
| Edit object | *In English grammar, the subject is the person, thing, or idea that performs the action of the verb in a sentence, while the object is the person, thing, or idea that receives the action of the verb. Generate all possible variations by changing only the object of the provided sentence. For example: Change "A man walking a dog" to "A man walking a horse". <caption>* |
| Edit background | *[leftmargin=*]Generate all possible variations of the provided sentence by changing or adding background or location details without altering the foreground or its attributes. For example: Change "A man walking a dog" and "A man walking a dog with mountains in the background", or change "A man walking a dog on grass". Change "A man walking a dog on the road" to "A man walking a dog with mountains by the beach". <caption> Generate all possible variations of the provided sentence by only changing the weather conditions, or adding a description of the weather if not already present. For example: change "A man walking a dog" to "A man walking a dog in the rain", and change "A man walking a dog in the rain" to "A man walking a dog in the snow". <caption>* |
| Edit domain | *Generate a few variations by only changing the data domain of the provided sentence without changing the content. For example: Change "A photo of a man" to "A painting of a man" or "A sketch of a man". A photo of <caption>* |
| Edit adjective | *Generate all possible variations of the provided sentence by only adding or altering a single adjective or attribute. For example: change "A man walking a brown dog" to "A tall man walking a brown dog", "A man walking a black dog" or "A man walking a cheerful brown dog". <caption>* |

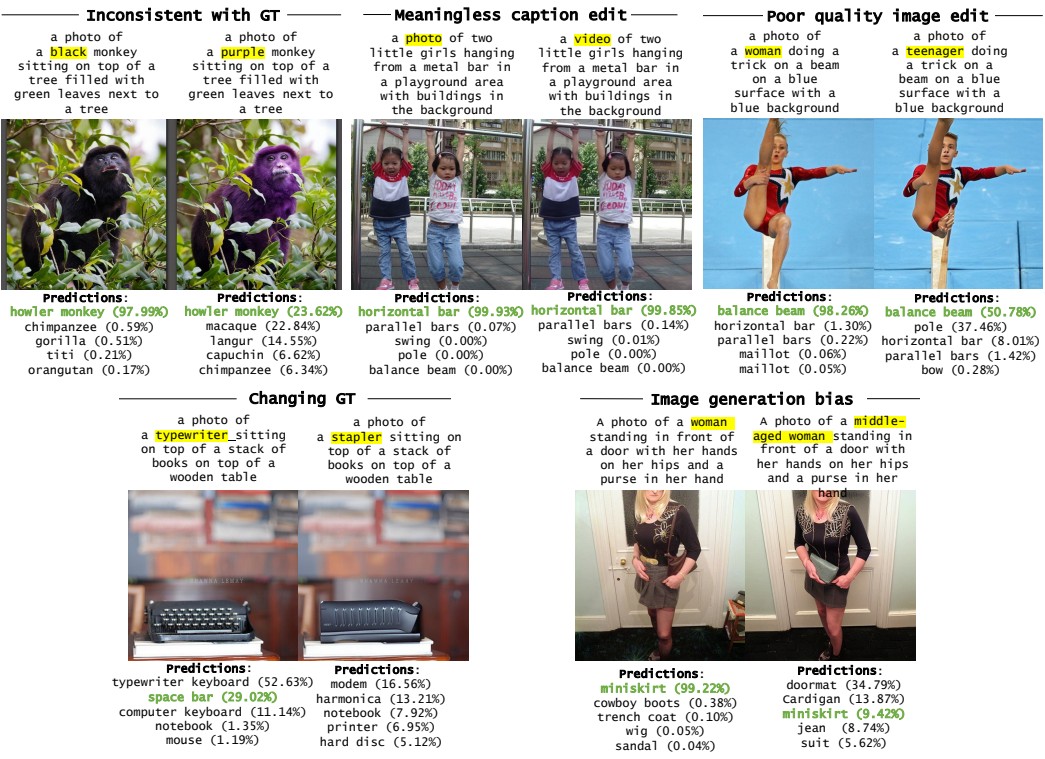

Figure 12: We visualize the different modes of failure of LANCE for genenerating counterfactuals on the HardImageNet dataset. Each row corresponds to a specific edit type. Above each image, we display its generated caption, highlighting the original and edited word(s). Below each image, we display the top-5 classes predicted by a ResNet-50 model and the associated model confidence, coloring the ground truth class in green.

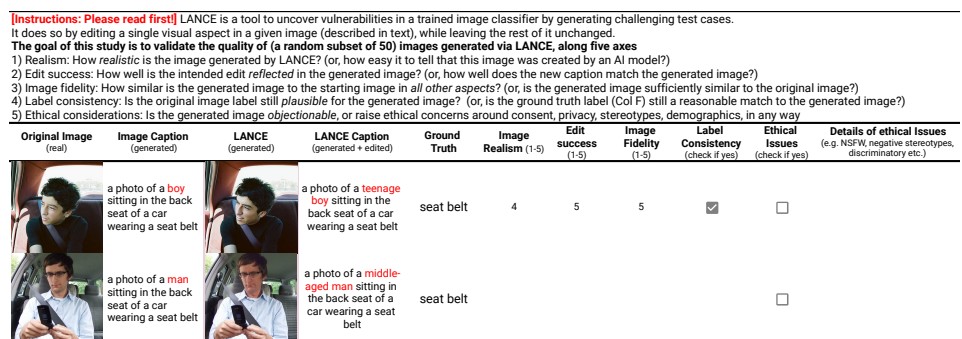

Figure 13: **Human study.** A screenshot of the interface we deploy to evaluate the quality of images generated by LANCE.

Table 6: Hyperparameter values used for caption (*top left*), LLAMA finetuning (*top right*) and image editing (*bottom*).

| config | value |
|---|---|
| min. caption length | 20 |
| max. caption length | 100 |
| repetition penalty | 1.0 |
| $\epsilon$ | 0.5 |
| decoding strategy | beam search |
| beam size | 5 |
| clustering strategy | K-Means |
| number of clusters | 5 |

(a) **Caption Generation and Clustering**

| config | value |
|---|---|
| learning rate | 3e-4 |
| effective batch size | 128 |
| grad. accumulation steps | 32 |
| weight decay | 0.0 |
| max. sequence len | 1024 |
| lora rank [54] | 8 |
| lora $\alpha$ [54] | 16 |
| lora dropout [54] | 0.05 |
| lora weights [54] | qkv |

(b) **LLAMA Finetuning**

| config | value |
|---|---|
| Stable Diffusion [16] version | 1.4 |
| image resolution | 512x512 |
| LR scheduler | DDIM scheduler [60] |
| `beta_start` [60] | 0.00085 |
| `beta_end` [60] | 0.012 |
| `beta_schedule` [60] | scaled linear |
| diffusion steps [19] | 50 |
| `attention_replace_edit` [19] | 2 |
| `cross_replace_steps` [19] | 0.8 |
| `self_replace_steps` [19] | $[0.4, 0.5, 0.6, 0.7, 0.8, 0.9]$ |
| guidance scale | 7.5 |
| $\tau$ [42] | 0.2 |
| img sim. threshold [42] | 0.7 |
| img-text sim. threshold [42] | 0.2 |

(c) **Image Editing**

