# OpenReview forum: "LANCE: Stress-testing Visual Models by Generating Language-guided Counterfactual Images"
_NeurIPS.cc/2023/Conference — NeurIPS 2023 poster_

### Official Review · Reviewer_Fgt7 · 2023-07-06

**Soundness:** 3 good
**Presentation:** 3 good
**Contribution:** 2 fair
**Rating:** 7
**Confidence:** 4

**Summary:**

This paper presents a method for stress-testing visual classifiers. The key idea is to use language to generate counterfactual images. Their approach can be summarized into four steps: 1) take an image input; 2) get its caption from a captioning model (BLIP-2); 3) perturb the caption by changing some words using ChatGPT and fine-tuned LLAMA; 4) generate new images based on perturbed captions using text inversion of Stable Diffusion. On the ImageNet dataset, they showed that various models such as ResNet-50 and ViT-B consistently achieve much lower performance on their generated new ImageNet compared to the original ImageNet, and the generated ImageNet enables surfacing model biases.

**Strengths:**

1. This paper targets a novel direction --- stress-test visual models, which is important in real-world vision applications.

2. The proposed method is technical sound, and the generated images from their pipeline look good and realistic.

3. The paper conducted experiments on the realistic ImageNet dataset instead of simple synthetic datasets, showing the practical values of their method in realistic scenarios.

4. This paper is generally well-written and easy to understand.

**Weaknesses:**

1. There is no baseline to compare, and the numbers are hard to justify their method's effectiveness. The main quantitive results from this paper showed that various vision models achieve much lower performance on their generated ImageNet dataset. However, it is unclear whether the generated images are 100% correct. If the text perturbation happens to change the class or introduce new classes to the image, the model prediction is expected to change and the performance will drop. Therefore, a human study showing the label validity of generated images is particularly necessary.

2. There are some existing works proposing very similar directions. For example, [1] is almost the same as this work, which uses textual reversion to convert an image into a token and compose this token with natural language and generate new images to test vision models. [2] uses off-the-shelf image captioning and generation models to discover model bugs. While [1] can be viewed as concurrent work, it would be helpful if the authors could discuss the similarity and differences between this work and [1-2].

3. From Figure 6 last row, changing the caption from "car" to "bus" generates almost the same images, but model predictions change a lot. The change may be because generated images have specific patterns that are not perceptible to humans, which cannot reflect the actual errors using real images.

4. The stress test will be limited by language itself, as language is hard to represent certain properties of objects, such as orientation and lighting. However, this is also pointed out by the authors and is minor considering the advantage of using language.

[1] Dataset interfaces: Diagnosing model failures using controllable counterfactual generation. https://arxiv.org/pdf/2302.07865.

[2] Discovering Bugs in Vision Models using Off-the-shelf Image Generation and Captioning. https://arxiv.org/pdf/2208.08831.

**Questions:**

- Algorithm 1:
  - Line 7 duplicate definition of "c" (previously used as the class, now used as the caption)
  - Line 13: ">" should be "<"

- Equation 2: "|" in the denominator should be "||"

- Line 282: what is the percentage for the generated counterfactuals to be semantically inconsistent with the ground truth label?

**Limitations:**

See weaknesses. I'm happy to increase my rating if these are well addressed, especially 1 and 2.

---

> ### Author Rebuttal · Authors · 2023-08-09
>
> Thank you for the thoughtful feedback!
>
> > … a human study showing the label validity of generated images is particularly necessary.
>
> Thanks for the suggestion! Please see our general response above for results from a human study evaluation of the quality of our dataset. Our generated dataset is adjudged to be high quality along several metrics, including image realism, edit success, fidelity, and ethics. Importantly, *respondents confirm label validity for 96.8% of generated images*.
>
> Also to clarify, we do propose and benchmark two additional baselines: performance on the image reconstruction from the latent diffusion model (''Reconstructed'' in Table 2), in order to verify DDIM inversion success, and a masked language-modeling based baseline (''LANCE-R'' in Table 2), both of which significantly lag behind LANCE.
>
> > It would be helpful if the authors could discuss the similarity and differences between this work and [1-2].
>
> Thanks for pointing these out!
>
> While our work and Wiles _et al._ [1] do share similarities (both use text-to-image and image-to-text models), they actually differ _significantly_ in both motivation and approach: Wiles _et al._ [1] uses the text description of a misclassified image as a cluster seed to generate additional images, and measures whether the cluster failure rate exceeds the baseline failure rate for that class. While this may discover failure modes with plausible rationales, these insights themselves might be less reliable: text-to-image models can generate a set of visually diverse images that satisfy a given prompt, which could each result in a misclassifcation for a different reason rather than due to a specific shared visual attribute.
>
> In contrast, LANCE generates counterfactuals via targeted editing (using prompt-to-prompt tuning with null-text inversion). By altering only a _specific_ visual attribute, we can be confident that a change in model confidence is indeed due to that change, and thus obtain more reliable insights. Further, by using each image from an IID test set as a potential probe rather than failure cases alone, LANCE can surface vulnerabilities even for classes with seemingly high performance (eg. for the ''sunglasses'' class, models performance is initially high but drops when the domain is changed to painting, see Fig. 6).
>
> While concurrent work by Vendrow _et al._ [2] claims to generate counterfactuals from user-defined (rather than automatically generated) variations, they adopt a more relaxed definition for counterfactual: for eg. consider the generated counterfactual examples highlighted in Fig. 5 of their paper, which differ from their original images in more ways than one. This is presumably due to learning a class-specific text token (via inversion) which is then used to generate counterfactuals. As described before, this can confound analysis, since images are altered in unpredictable ways. In contrast, our approach results in significantly better edits (eg. see Fig 5 in our paper and Fig 4 in supp.), and therefore yield more reliable insights into model vulnerabilities.
>
> [1] Wiles et al., Discovering Bugs in Vision Models using Off-the-shelf Image Generation and Captioning. https://arxiv.org/pdf/2208.08831.
>
> [2] Vendrow et al., Dataset interfaces: Diagnosing model failures using controllable counterfactual generation. https://arxiv.org/pdf/2302.07865.
>
> > The change may be because generated images have specific patterns that are not perceptible to humans, which cannot reflect the actual errors using real images.
>
> LANCE surfaces a diverse range of model failure modes, some of which are highly interpretable (eg. sensitivity to background color for the ''howler monkey'' class), while others may be less so (as correctly pointed out by the reviewer). We include examples of both to highlight the diversity of discovered failure modes; admittedly, not all lead to actionable insights.
>
> Thank you for pointing out errors in the text, we will incorporate all suggestions!

---

> > ### Comment · Reviewer_Fgt7 · 2023-08-14
> > **Thanks for the reaponse**
> >
> > We thank the authors for the detailed response. The answers to most questions seem persuasive, therefore I'm increasing my score.
> >
> > Additional question: In response PDF Figure 1, it seems both edits are not successful (at least I cannot see there is any semantic change between the original images and edited images, e.g., from "boy" to "teenage boy", from "man" to "middle-aged man"), but it was rated as success (rated 5 in the range of 1-5). I'm worried about the data annotation quality.

---

> > > ### Author Response · Authors · 2023-08-16
> > > **Thank you for considering our rebuttal!**
> > >
> > > We appreciate the reviewer's effort in considering our response and are glad it was able to address their concerns.
> > >
> > > In both generated images on PDF Figure 1, the person's facial features are modified to make them look older (best-viewed zoomed-in). Further, the screenshot included in the screenshot is only from a single respondent. On average, the average edit success rating (mean and 1 standard deviation) for the first edit was 4.27$\pm$0.65, and 4.82$\pm$0.40 for the second, indicating that at least a few respondents found both edits to only be partially successful.
> > >
> > > We took several measures to ensure that our human study was high quality, including selecting external respondents who were uninvolved in the project in any capacity and providing carefully-worded instructions. In fact, the average response time recorded was about 20 minutes for the full study or $\sim$24 seconds per question, which we believe is adequate time to provide carefully considered responses.
> > >
> > > We hope this addresses the reviewer's concern, and would be happy to provide any further clarifications.

---

> > > > ### Comment · Reviewer_Fgt7 · 2023-08-18
> > > > **Thanks for the additional explanation**
> > > >
> > > > The clarification on the examples in the rebuttal PDF makes sense to me. I increase my score to 7 regarding this.

---

### Official Review · Reviewer_9YtG · 2023-07-06

**Soundness:** 3 good
**Presentation:** 3 good
**Contribution:** 3 good
**Rating:** 5
**Confidence:** 2

**Summary:**

The paper proposes a way to generate new data to evaluated the robustness of model. It first captions an image, then changes part of it, then generates an image from the modified caption. The model is then evaluated on these new images to see how robust they are to things like background changes.

**Strengths:**

The idea is interesting and is an easy way to generate new evaluation data. The paper is well written and easy to follow. The approach would be a valuable tool to evaluate models.

**Weaknesses:**

Since it is an automated benchmark, it is unclear how good the quality is. It relies on a lot of parts working right (captioning model, caption changing model and image generation model).  It would be good to have some human evaluation in Table 2 to see how well humans do on this set.

**Questions:**

How can the quality of the generated dataset be evaluated? How do we know it is generating a good benchmark?

**Limitations:**

Yes.

---

> ### Author Rebuttal · Authors · 2023-08-09
>
> Thank you for the thoughtful review, and for recognizing the potential of LANCE as a valuable diagnosis tool!
>
> > It would be good to have some human evaluation in Table 2 to see how well humans do on this set. How can the quality of the generated dataset be evaluated? How do we know it is generating a good benchmark?
>
> Please see our general response above for results from a human study evaluation of the quality of the generated dataset, as well as quantitative evaluations of each component of our method.
> Our dataset is adjudged to be high quality (via a human study) in realism, edit success, and fidelity. Additionally, we manually inspect all images in the dataset and filter out images that are potentially unethical.
>
> While measuring human accuracy in Table 2 is a great suggestion, having humans perform 1000-way classification is challenging;  as a proxy, we instead measure the label _consistency_ of our generated images, to ensure that the original ground truth label is still valid: encouragingly, our respondents find this is the case for 96.8% of images on average.

---

> > ### Author Response · Authors · 2023-08-18
> > **Thank you for the thoughtful review!**
> >
> > We sincerely hope that our rebuttal has addressed the reviewer's concerns, and would be happy to answer any lingering questions. Otherwise, we would appreciate if they would consider updating their score in light of our response.

---

### Official Review · Reviewer_muqu · 2023-07-14

**Soundness:** 3 good
**Presentation:** 3 good
**Contribution:** 3 good
**Rating:** 6
**Confidence:** 2

**Summary:**

This paper proposed a new testing protocol LANCE for existing vision models.
It adopts LLM-based caption perturbation and DDPM based image editor to generate counterfactual samples.
Specifically, for each image, LANCE first use a pre-trained captioner (BLIP-2) to generate the corresponding caption.
Then it adopts an LLM to perturb the caption with different factors (subject, object, background, etc.).
The goal of perturbation is to generate a caption corresponding to a counterfactual image with semantics preserved.
With the perturbed caption, the author generates another image with counterfactual aspects and similar looking with the original image.
The authors observed a significant performance drop when testing image classification models on these generated images.
Furthermore, the authors also present a detailed analysis based on the perturbation factors.

**Strengths:**

1. The authors establish an interesting pipeline to generate counterfactual hard examples for testing vision models. It involve an LLM-based image caption perturbator and a ddpm-based image editor. Qualitative results show that this pipeline can generate some impressive counterfactual images.
2. The authors present a detailed analysis on the performance degradation when testing on the generated counterfactual images.

**Weaknesses:**

1. The authors should present more quantitative statistics about the counterfactual examples generation pipeline, for example:
    1. What is the success rate of an LLM to generate a plausible perturbed caption (GPT-3.5 & Finetuned LLAMA-7B). Also, I noticed that you are using Nvidia A40 (48GB), thus a comparison between different LLAMA variants (+13B, e.g.) are also welcomed.
    2. What is the success rate of DDPM-based image editing? I'm specially interested in that since I observe poor quality when using StableDiffusion to generate common images (compared to other advanced AIGC models).
2. Are all newly generated images (781 images, according to the supp) verified manually by human? If so, how many worker-hours are required to generate these samples?

**Questions:**

See Weakness.

**Limitations:**

See Weakness.

---

> ### Author Rebuttal · Authors · 2023-08-09
>
> Thank you for the thoughtful review!
>
> > What is the success rate of an LLM to generate a plausible perturbed caption
>
> Please see our general response above for a quantitative evaluation of our caption perturbation strategy: to summarize, we find the perturber to be 89% accurate at following the prompt provided (i.e. _correctly_ editing _only_ the domain if instructed to do so). We note that even in failure cases (eg. additionally changing an irrelevant word), the perturbed captions are almost always _plausible_. Further, edits that mutate the ground truth are correctly filtered out 98.4% of the time.
>
> > What is the success rate of DDPM-based image editing?
>
> As our general response shows, our generated dataset is adjudged to be high quality (via a human study) in realism, edit success, and fidelity. Respondents also confirm label validity for 96.8% of generated images. We note that generating high-quality edits for ImageNet-style images with Stable Diffusion requires several additional measures that we describe in Sec. 3.2, such as using prompt-to-prompt with null-text inversion and (automated) hyperparameter tuning. That said, we emphasize that LANCE is not tied to a specific text-to-image model and can easily be updated to include more recent generative models.
>
> > Are all newly generated images (781 images, according to the supp) verified manually by human? If so, how many worker-hours are required to generate these samples?
>
> Yes. As detailed in our general response above, our generated images are adjudged via a human study (11 respondents, 15-20 minutes per response) to have a high image and edit quality and fidelity, while almost always preserving the ground truth label. Further, we also manually screen all 781 images for ethical and quality issues (which took approximately 1 hour), in the process flagging 8 images that we filter out.
>
> > A comparison between different LLAMA variants (+13B, e.g.) are also welcomed.
>
> We use zero-shot prompting (using the same prompts as before, see Table 1 in supp.) with the recently released LLAMA-2-13B chat model. We find that even without any task-specific finetuning, it leads to high-quality text edits, for eg:
>
> **LLAMA-2 Prompt**: Generate all possible variations of the provided sentence by only changing the weather conditions to a _rare celestial event_: "a photo of a red umbrella sitting on top of a lush green field next to a sidewalk next to a building"
>
> **Response:**
> 1. "a photo of a red umbrella sitting on top of a lush green field next to a sidewalk next to a building under a lunar rainbow."
> 2. "a photo of a red umbrella sitting on top of a lush green field next to a sidewalk next to a building during a meteor shower."
>
> ...
>
> 10. "a photo of a red umbrella sitting on top of a lush green field next to a sidewalk next to a building during a rare solar eclipse."
>
> Besides obviating additional finetuning, this also opens up the possibility of using LANCE to stress-test against custom variations beyond the ones described in this paper.

---

> > ### Author Response · Authors · 2023-08-18
> > **Thank you for the thoughtful review!**
> >
> > We sincerely hope that our rebuttal has addressed the reviewer's concerns, and would be happy to answer any lingering questions. Otherwise, we would appreciate if they would consider updating their score in light of our response.

---

### Official Review · Reviewer_G3fV · 2023-07-19

**Soundness:** 3 good
**Presentation:** 4 excellent
**Contribution:** 2 fair
**Rating:** 5
**Confidence:** 4

**Summary:**

The paper proposes to use recent advances in text-to-image generation/editing to design evaluation by generating "counterfactual" data. The main idea of the method is to make semantically irrelevant (to the output class) edits to an image and observe its effects on the model predictions. A variety of models including CNNs, and ViT (trained classically as well as via language supervision) are tested, which show that such editing operation can indeed make a "hard" test set where most tested methods struggle to do well.

**Strengths:**

[S1] Great Presentation Clarity: The paper is very well written and the details about what has gone into the final data-generation scheme are presented in sufficient detail. The experimental setup, the algorithm, and the implementation of the baseline models are clearly explained and should be easily reproducible independently.


[S2] Good evaluation/results: The proposed model, based on the experiments presented, show a huge potential for using counterfactual generation of images to stress-test models. While the paper only makes a handful of edit types, there is a possibility to design hyper-specific probes (does change clothing color change models predictions about a person's profession?) that can be designed for specific applications.

**Weaknesses:**

[W1] Need more discussion of scope and their limitations: While the main idea is sound and the descriptions are clear, the evaluation/discussion leaves a lot to be desired. Most of the evaluation is done "end-to-end" which does not give much insight into the success of the various stages involved in the process. Some example concerns that I have:

- The success of checks and balances (L155 and L185) is not verified/discussed. Do they always work? When do they fail? What are the effects of their failing?
- What is the diagnostic prowess of this method? Besides showing that the edit-based perturbations does indeed make models flip their predictions (which has also been previously shown via adverserial noise injection), what insights can we draw from this type of stress testing? Does the results of this type of probe correlate with other tests (perhaps for bias/fairness and/or other diagnostic datasets? Do the edits represent (in the paper's view), the realm of possible naturally occurring variations in background/unrelated content in the picture? If not, can this still diagnose/predict the model's robustness in the real world? Or, is this just another form of discovering "adversarial attack" on the model predictions?

There is only a very short limitation section at the end of the conclusion section. I would have liked to see a more open discussion about exactly what (in view of the paper) can this method tell us about a model and what it cannot.


[W2] No manual/external/independent validation: The quality of the edits, and the accuracy of the labels post-edit are not validated by any external/manual evaluation. We only see decreased overall accuracies for popular methods. While a simple baseline of random edit is tested, it is not clear whether a noise perturbation/other simpler perturbation techniques could also result in similar loss. Regardless of the loss in accuracy produced, it is also important to know whether the editing process leaves any unwanted artifacts. E.g., In the headline image, there is no actual sled visible, and the prediction of it likely comes from context. As such, it is unsurprising that changing the dog breed to pomeranian (an unlikely breed to be pulling sleds, also both "husky" and "pomeranian" are actual categories by themselves) would result in flipped predictions. To be clear, I am not claiming that this is "OKAY" behavior, just that perhaps the mechanism is "label noise" in Imagenet rather than "context bias" (Since there is nothing but the context that can help us make predictions). I am claiming, however, that this is a different category of error than other examples in the paper. Both are important, and the proposed method is likely to help uncover both, but at a lack of external validation about the accuracy of the labels before and after the edit operation. Some questions that I'd have liked validated externally:
- Can the accuracy of the "perturber" be measured independently?
- What is the accuracy of the ground truth labels post-edit? Are they truly unchanged? Are the methods presented in L185 and L155 (re. checks and balances) enough? Do they work? Have they avoided this issue successfully?
- Since there is a pervasive error in Imagenet (https://arxiv.org/abs/2103.14749), does this method have any correlation with the noisy labels? In other words, is this intervention making "easy to predict" examples harder or primarily just the "confusing" ones?

**Questions:**

Please see the weaknesses above

**Limitations:**

Please see the weaknesses above

---

> ### Author Rebuttal · Authors · 2023-08-09
>
> Thank you for the thorough review! In our general response, we evaluate each component of our method (including checks and balances), and include a human study to assess the overall realism, edit success, and fidelity of our generated dataset.
>
> > Can the accuracy of the ''perturber'' be measured independently?
>
> Yes! We measure the accuracy of the perturber at following the provided instruction: specifically at making the type of edit specified while leaving the remainder of the caption unaltered. We observe an overall accuracy of 89% (see general response for details).
>
> > What is the accuracy of the ground truth labels post-edit?
>
> In our human study, respondents confirm label validity for 96.8% of generated images, certifying that LANCE-generated images almost never mutate the ground truth label.
>
> > The success of checks and balances (L155 and L185) is not verified/discussed. Do they always work? When do they fail?
>
> Image editing: See previous response.
>
> Caption editing: We measure the robustness of our strategy to filter out caption edits that mutate the ground truth, and find that our method successfully filters out 98.4% of such edits (see general response for details).
>
> > I would have liked to see a more open discussion about exactly what can this method tell us about a model and what it cannot.
>
> Sorry about that! We actually included a characterization of LANCE’s failure modes in supp. (Fig. 5), and in our general response above include an expanded discussion of the limitations of our method.
>
> >  What is the diagnostic prowess of this method?
>
> **Class-level insights**. In Section 4.2 (L261), we show how LANCE can provide class-level insights by clustering edits and visualizing clusters with high sensitivity ($\Delta p(y_{GT}|\mathbf{x})$, Eq. 4). Via this we discover, for instance, that for the ''sunglasses'' class, ResNet-50 models are highly sensitive to the image domain, and underperform on renditions of the same images as paintings. In Sec. 2 of supp. (L28), we extend this analysis to other classes and find that accuracy of a ResNet-50 model drops consistently for the photo$\to$painting across classes.
>
> **Bias Diagnosis**. Some of LANCE’s discovered error modes are indeed corroborated by other diagnostic datasets, eg. models have been shown (by collecting segmentation masks) to rely heavily on class-specific spurious cues to make predictions on the 15 classes included in the HardImageNet benchmark. LANCE discovers several of these without additional labels: eg. discovering that for the ''howler monkey'', model predictions are sensitive to the color of the surrounding foliage (Figure 6, top row).
>
> **Beyond conventional diagnosis**. LANCE can surface nuanced error modes that conventional diagnostic datasets that are constrained to an IID test set cannot. Our teaser figure is a good example, wherein LANCE surfaces a model reliance on dog breeds to predict ''dog sled''. We agree with the reviewer that whether such reliance is acceptable may be breed-dependent. In the rebuttal PDF, we investigate if the model confidently predicts ''dog sled'' for two other popular sled-dog breeds (Malamutes and Samoyeds), and find that it frequently does not! This suggests that the model strongly associates the ''dog sled'' class with huskies.
>
> > While a simple baseline of random edit is tested, it is not clear whether a noise perturbation/other simpler perturbation techniques could also result in similar loss. Does the editing process leave any unwanted artifacts?
>
> To clarify, our random edit baseline only picks the word to edit at random but uses a masked language model to make the actual edit. To ensure that the editing process does not leave unwanted artifacts, we: i) validate that accuracy over reconstructed images (with the original prompt) is preserved. ii) only retain edits with high CLIP directional similarity (Eq. 2). The high ratings for image realism obtained in our human study further attest to the generation quality.
>
> > Do the edits represent (in the paper's view), the realm of possible naturally occurring variations in background/unrelated content in the picture? If not, can this still diagnose/predict the model's robustness in the real world?
>
> Our dataset contains >160 unique weather patterns, spanning generic (eg. windy, snowy, rainy, sunny, misty, hazy, blizzard, thunderstorm, hailstorm, humid) and specific (eg. bitterly cold wind, sunny day in the springtime, cold winter morning, blustery winter solstice) types. Similarly, it contains >190 unique backgrounds, including generic (eg. park, lake, ocean, river, swimming pool, beach, scenic mountain range, city street) and specific ones (deserted stadium on a rainy day, a luxurious hotel, small oceanarium).
>
> Further, we find that with recent LLMs (eg. LLAMA-2 13B chat), we can generate challenging background edits (eg. ''rare celestial events'') with _no additional finetuning_! This opens up the possibility of using LANCE to stress-test against custom variations beyond the ones described in this paper.
>
> > Is this just another form of discovering "adversarial attack" on the model predictions?
>
> No, we do not consider LANCE as an adversarial attack on model predictions, since by design, it is not optimized end-to-end to fool a downstream classifier, and interventions are only made through language. We will clarify.
>
> > Does this method have any correlation with the noisy labels?
>
> Great question! To investigate, we cross-reference labeling errors found in Northcutt _et al._ [A] for the ''howler monkey'' class in ImageNet. They find only a single mislabeled example (true label: ''white-headed capuchin''), whereas we discover a consistent failure mode (reliance on background color). Further, in Table 2 we also report accuracy@5, which is less sensitive to changes in predictive order when multiple labels are plausible, and find that LANCE-generated test sets also lead to larger drops in accuracy@5 (eg. 3% for ResNet-50).

---

> > ### Author Response · Authors · 2023-08-18
> > **Thank you for the thoughtful review!**
> >
> > We sincerely hope that our rebuttal has addressed the reviewer's concerns, and would be happy to answer any lingering questions. Otherwise, we would appreciate if they would consider updating their score in light of our response.

---

### Author Rebuttal · Authors · 2023-08-09

We thank reviewers for their thoughtful feedback. We are delighted that they found our problem novel and broadly applicable (reviewers Fgt7, G3fV), our method valuable and technically sound (reviewers muqu, 9YtG), our experiments realistic and reproducible (reviewers muqu, Fgt7, G3fV), our results impressive (reviewers muqu, Fgt7), and our paper very well-written (reviewers G3fV, 9YtG, Fgt7). We also appreciate and will incorporate the feedback from our ethics reviewers. **We will make our dataset publicly available as requested**.

The primary concern shared by reviewers pertains to the limited evaluation of LANCE’s components and an inadequate discussion of its limitations and ethical implications.

We take these concerns seriously, and describe below our efforts to address them.

1) **Evaluation**. We independently evaluate (**full tables in PDF**):


    a) **Caption perturber**. In Fig. 4 of the paper, we verified caption realism by reporting the low average perplexity of the generated and perturbed captions. We now also evaluate the accuracy of the perturber at making the type of edit specified by the prompt (eg. at _correctly_ changing _only_ the domain for a domain edit). To do so, we manually validate a random subset of 500 captions and their perturbations (100 per-type) generated by LANCE. We observe an overall accuracy of 89%, with the _object_ (84%) and _adjective_ (82%) types achieving the lowest accuracies: we find that most failures for these types result from incorrect identification of the object or adjective in the sentence. We note that even in failure cases (eg. also changing an irrelevant word), the edits made are still reasonable.

    Next, we evaluate the efficacy of the checks and balances we insert to weed out caption edits that alter the label. We first label the same subset of 500 examples by hand with the correct action (filter / no filter), and compare against our filtering strategy. Across perturbation types, our filtering strategy achieves both high precision and recall, with a low overall false positive (2.8%) and false negative (1.6%) rate. Of these, a majority of false negatives (edits that alter the ground truth that we fail to catch) are for the _subject_ and _object_ types: we find these typically change the ground truth inadvertently. For example, for an image of a typewriter labeled ''keyboard space bar'', the edit typewriter$\to$painting erroneously passes our filter but mutates the ground truth, as paintings do not usually have space bars.


    b) **Image generator.** In Fig. 4., we verify the realism of the generated images by reporting FID scores. As recommended by reviewers, we now conduct a human study to measure performance along 5 axes:

    * **Image Realism** (1-5, 5=best): How easy is it to tell that the counterfactual image was generated by AI?

    * **Edit success** (1-5, 5=best): Does the generated image correctly incorporate the edit?

    * **Image fidelity** (1-5, 5=best): Are all the changes made relevant towards accomplishing the edit?

    * **Label consistency** (Yes/No): Is the original image label still plausible for the generated image?

    * **Ethical issues** (Yes/No + Text box): Is the generated image objectionable, or raise ethical concerns around consent, privacy, stereotypes, demographics, etc.?

    We collect responses from 11 external respondents for a random subset of 50 <image, ground truth, generated caption, perturbed caption, counterfactual image> tuples (10 per-type), and report mean and variance. See the rebuttal PDF for a screenshot of the interface and full tabulated results.

    Images generated by LANCE are rated to be high on realism (4.3/5 on average)  and fidelity (4.6/5 on average), and slightly lower on edit quality (3.8/5 on average). Of these, we find that background edits score lowest on edit success, due to sometimes altering the wrong region of the image. Further, generated images have high label consistency (96.8%), indicating that our checks and balances are effective at preserving the ground truth label. Finally, <2% of images are found to raise ethical concerns: we discuss these in detail below.



2) **Ethics and limitations.** On inspection of the images marked as objectionable in our human study, we find that most arise from bias encoded in the generative model: eg. for the edit woman$\to$lifeguard, the generative model also alters the perceived gender and makes the individual muscular. Further, for people$\to$athlete, the model also alters the person’s race. The model also sometimes imbues stereotypical definitions of subjective characteristics such as ''stylish'' (eg. by adding makeup). To address this, we manually inspect the entire dataset of 781 images for similar issues and exclude 8 additional images. Going forward, we hope to address this in the generation pipeline itself, by excluding inherently subjective text edits, and by using recent text-to-image with improved fairness and steerability.

    **Limitations.** In Fig. 5 of supp., we categorized and visualized the most common failure modes of our approach (re-included in rebuttal PDF). We enumerate them below:

    a) Label-incompatible text edit: eg. black howler monkey$\to$purple howler monkey (howler monkeys are almost always black or red).

    b) Poor quality image edit: Images with low realism, which are largely eliminated via our checks and balances.

    c) Image edit that inadvertently changes the ground truth: eg. typewriter$\to$stapler for the class ``space bar’’ (staplers do not contain space bars).

    d) Inherited bias: eg. changing perceived gender based on stereotypes.

    e) Meaningless caption edit: eg. boy in back seat $\to$ driver in the back seat.

    Such failures can confound an observed performance drop over a LANCE-generated counterfactual image. However, we observe such failures rarely, and find that class-level and aggregate performance variations still lead yield reliable insights.

---

### Decision · Program_Chairs · 2023-09-21

**Decision:**

Accept (poster)

**Comment:**

After discussion with the authors, all reviewers recommend acceptance to varying degrees. The AC does not see sufficient grounds to overturn this consensus.

It is important to note that this outcome is in part the result of three human studies presented during the discussion period that support the efficacy of two submodules. These studies are relatively small scale (largest at 11 participants), were presumably performed in a short time period, and two of the three rely on authors as subjects. From the AC's perspective, these therefore represent promising preliminary results rather than rigorous evaluations. Given the other merits of the work, these are still useful for building trust about the methodology. Authors are encouraged to strengthen these studies if possible and be clear about their limitations in the camera ready.

The discussion of limitations presented in the overall response and the rebuttal are useful. The AC believes these should be elevated to the main paper if possible. Likewise, the discussion with reviewer G3fV of precisely what diagnostic scenarios LANCE is likely to enable is useful context for readers. Discussion of these and their potential interactions with the method's limitations should be included.